

# Multi-Point In-Situ Measurements of Turbulent Flow in a Wind Turbine Wake and Inflow with a Fleet of UAS

Tamino Wetz[1] and Norman Wildmann[1]

[1]Deutsches Zentrum für Luft- und Raumfahrt e.V., Institut für Physik der Atmosphäre, Oberpfaffenhofen, Germany
**Correspondence:** Tamino Wetz (tamino.wetz@dlr.de)

**Abstract.** The demand on wind energy for power generation will increase significantly in the next decade due to the transformation towards renewable energy production. In order to optimize the power generation of a wind farm, it is crucial to understand the flow in the wind turbine wake. The flow in the near wake close downstream of the wind turbine (WT) is complex and highly three-dimensional. In the present study, for the first time, the SWUF-3D fleet of multirotor UAS is deployed for field

measurements on an operating 2 MW WT in complex terrain. The UAS-fleet has the potential to fill the meteorological gap of observations in the near wake with high temporal and spatial resolution wind vector measurements plus temperature, humidity and pressure. During the experiment, the flow up- and downstream of the WT is measured simultaneously. Various flight patterns are used to investigate the near wake of the WT. The velocity deficit and the turbulence profile in different downstream distances are measured by distributed UAS which are aligned perpendicular to the flow in the near wake. The results show

the expected double-Gaussian shape in the near wake under nearly stable atmospheric conditions. However, measurements in unstable atmospheric conditions with high turbulence intensity levels lead to single Gaussian-like profiles at equal downstream distances ($< 1D$). Additionally, horizontal momentum fluxes and turbulence spectra are analyzed. The turbulence spectra of the wind measurement at the edge of the wake could reveal that tip vortices can be observed with the UAS.

Keywords: atmospheric boundary layer, wake flow, turbulence, wind, multicopter UAS.

## 1 Introduction

According to statements by the International Energy Agency (IEA) (2022), the role of renewable energies in power generation will increase dramatically over the next decade. In the Announced Pledges Scenario (APS), renewable energy will outpace fossil fuels for electricity generation by 2030. This transformation is necessary to achieve the goal of net zero emissions in

2050. An annual increase of $18\,\%$ in wind energy capacity is needed to reach the next milestone in 2030 with global wind power generation of 7300 TWh (IEA, 2022). On the one hand, an increased demand for wind energy is met by optimizing the power output of a individual turbine, which is mainly achieved with a further enlargement of the WT rotors. On the other hand, the power output of entire wind farms must be optimized. The wake that forms behind a wind turbine plays a central role in the optimization of wind farm design and control. The wakes significantly reduce the power output of the downstream

turbines when operating in the wake. The increased size of WT leads to even more pronounced wakes and shorter relative





distances between turbines, especially in wind farms that undergo repowering. In order to cover the enormous demand for wind energy, the available space will have to be used as efficiently as possible. This requires closer staggering in wind farms, which in turn leads to more significant wake effects. Veers et al. (2022, 2019) mention the understanding of turbulent flow around WTs in the atmospheric boundary layer as one of the major challenges in wind energy research, which is supported by

Porté-Agel et al. (2019). This need for research is caused by the high complexity of the atmospheric flow and its interaction with the WT. The wake, which forms downstream of a WT, particularly causes losses on the downstream WT in wind parks due to the velocity deficit (Sanderse et al., 2011). Additionally, the increased turbulence can lead to higher fatigue loads on the downstream turbines (Frandsen, 2007).

In order to understand and classify the wake measurements, a brief overview of the flow around a WT is given below.

In general, the flow in the atmospheric boundary layer is affected both upstream and downstream of the WT. The induction zone is the area upstream in which the wind speed is reduced due to blockage effects from the WT. The velocity reduction can be estimated depending on the upstream position and the induction factor (Simley et al., 2016). The regime downstream of the turbine is divided into a near wake and a far wake (Vermeer et al., 2003). The near wake is characterized by highly heterogeneous and complex flow distribution and is closely related to the design of the WT. This region extends from the rotor

plane to a distance of 2 to 4 rotor diameter $D$ downstream (Wu and Porté-Agel, 2012). The flow field of the far wake is less heterogeneous, resulting in a more universal velocity deficit distribution and is less influenced by the detailed design of the WT. Further downstream, the velocity increases continuously towards free-stream velocity while the turbulence intensity decreases (Porté-Agel et al., 2019). The wake recovery is found to be complete at distances up to $8\,D$ downstream, but extend further for large rotor diameters and under stable atmospheric conditions where wakes persist much longer than in unstable conditions

(Fuertes et al., 2018). In this study we focus on field experiments in the near wake. The most prominent flow structure in the near wake is the tip vortex helix, besides the root vortex of the blades and the hub vortex. The helical vortex structure results from tip vortices which are induced by the pressure difference of the pressure and the suction side at the rotor blade tips. Various research groups studied the formation and the stability of tip vortices (Sherry et al., 2013; Zhang et al., 2011; Odemark and Fransson, 2013). The formation and the stability of tip vortices are of great interests due to the fact that tip

vortices prevent the outer flow entrainment into the near wake (Lignarolo et al., 2014). Therefore, the breakdown of the helix structure enhances turbulent mixing from the outer flow towards the wake center which leads to wake recovery. This area is typically characterized as the transition between near and far wake, where the mixing from outside towards the center of the wake significantly increases (Wu and Porté-Agel, 2012). The breakdown of the helix usually starts when helix vortex pairing occurs (Odemark and Fransson, 2013). In convective conditions the lifetime of tip vortices is significantly reduced by the high

turbulence in the free-stream (Lu and Porté-Agel, 2011).

Due to conservation of momentum the rotation of the WT induces a flow deflection in the opposite direction, such that for clockwise rotating WT (viewed from upstream towards the WT) the wake rotates counterclockwise (Manwell, 2009). This rotation is distributed over the entire rotor swept area and decreases further downstream. Depending on the yaw-misalignment of the WT (Bastankhah and Porté-Agel, 2016), the atmospheric conditions (Abkar and Porté-Agel, 2015) and the direction of

WT rotation (Englberger et al., 2020) different wake patterns can develop further downstream.



In the far wake the wind speed deficit is often modeled as a Gaussian distribution around the center, implying the lowest velocity in the center of the WT. However, in the near wake the averaged wind deficit can also be distributed in a double-Gaussian shape in lateral direction indicating high velocities around the center of the wake while the wake edges are characterized by regions of low velocities. The high velocity region in the center is due to low momentum extraction by the blades at small radius (Magnusson, 1999; Crespo et al., 1999; Keane et al., 2016; Bastankhah and Porté-Agel, 2017; Krogstad and Adaramola, 2011; Machefaux et al., 2015). Detailed reviews about wake aerodynamics are conducted by Porté-Agel et al. (2019); Vermeer et al. (2003).

In the past, the wake of WTs has been extensively studied using numerical methods, from basic engineering analytic models for single turbines (Jensen, 1983), wind farm optimization (Bastankhah and Porté-Agel, 2014) towards high-resolution large eddy simulations (LES) (Mehta et al., 2014). The flow around WTs has been examined in great detail in wind tunnel experiments by many research groups using various flow measurements techniques such as particle image velocimetry (PIV) (Sherry et al., 2013; Bastankhah and Porté-Agel, 2017) or flow visualization with smoke (Alfredsson and Dahlberg, 1979). In addition to the laboratory flow, the wake is studied in field campaigns under real atmospheric flow conditions. Most prominent in the last decades are measurements by remote sensing technologies such as Doppler wind lidar. Lidar measurements are carried out both from the ground and directly from the nacelle of a WT. Ground-based lidar measurements are used, for example, to investigate the wake characteristics and development in complex terrain (Wildmann et al., 2020; Menke et al., 2018). They can be used to determine the wake-center and track the extension of the wake in the far wake (Wildmann et al., 2018b), analyse turbulence within the wake (Wildmann et al., 2020) or study the wake length dependency on atmospheric conditions Wildmann et al. (2018a). Nacelle lidars or even spinner-integrated lidars are often used for flow measurements to optimize the active yaw control of the WT (Mikkelsen et al., 2012). In addition, they are conducted for characterizing the WT wake (Aitken and Lundquist, 2014; Brugger et al., 2020; Machefaux et al., 2015; Fuertes et al., 2018) and for model validation (Doubrawa et al., 2020).

More qualitative studies on wake flow characteristics, including coherent structure analysis, are performed by Yang et al. (2016); Abraham et al. (2021); Dasari et al. (2018) using snowflakes to visualize coherent structures. Even PIV was implemented by Abraham et al. (2021) employing snowflakes as a tracer to determine wind speed.

In addition to remote sensing, in-situ measurements were carried out to study the flow around WT. Airborne systems that were used for this purpose range from uncrewed flight systems (UAS) in fixed-wing configurations (Kocer et al., 2011; Wildmann et al., 2014; Reuder et al., 2016; Mauz et al., 2019; Alaoui-Sosse et al., 2022) and rotary wing configurations (Thielicke et al., 2021; Li et al., 2022) to crewed measurements around wind parks with a Dornier DO-128 research aircraft (Platis et al., 2021). All the mentioned in-situ measurements are based on only one device. This allows only a single time step at a variable spatial position, or (for multicopters) a single time series at a fixed spatial position . The approach of using a fleet of multicopters enables highly resolved observations at multiple spatial positions simultaneously. Besides the simultaneous measurement of inflow and wakes, it is possible to conduct multiple time series of the flow at different discrete positions in the wake.





The objective of this work is to examine the near wake of the WT in operational conditions and can be divided into different hypotheses and research questions:

– Can a fleet of UAS measure a double Gaussian velocity deficit and turbulence intensity profile in the near wake of a WT?

– Do the horizontal momentum fluxes point towards the inner wake at the edge of the wake?

– It is possible to capture the tip vortex with multicopter measurements at the edge of the wake?

– Do the near wake characteristics significantly changes in different downstream distances ($< 2D$)?

– What are the influences of atmospheric stability on the near wake regarding the velocity deficit and the turbulence intensity?

The present study is structured as follows: First, in Sect. 2, we describe the experimental setup, including the UAS fleet, the measurement location, and the flight strategies. Various methods that are necessary for the evaluation and discussion of the
data are then explained in Sect. 3. The results of the field measurements of the UAS fleet on a WT are presented in Sect. 4 and then discussed (Sect. 5). Finally, the results are summarized in a conclusion (Sect. 6).

## 2 Experiment

### 2.1 Measurement Hardware

The SWUF-3D (Simultaneous Wind measurement with Unmanned Flight Systems in 3D) fleet consists of more than 30 quadro-
tor UAS. The dimensions of the UAS are relatively small with a distance between two rotor axes of 0.25 m and a take-off weight of 0.645 kg. The UAS is controlled by an autopilot based on inertial measurement unit (IMU) and global navigation satellite system (GNSS) data. The wind speed is measured in hover flight without an additional flow sensor. The wind measurements are carried out by relating the quadrotor movements to the acting wind forces while hovering at a fixed position. With this method accurate wind speed and wind direction measurements can be achieved for the entire fleet (Wetz and Wildmann, 2022).
In addition, temperature and humidity are measured by an external sensor. The hardware and the wind algorithm are described in more detail in earlier publications (Wetz et al., 2021; Wetz and Wildmann, 2022). There we showed that turbulent structures can be resolved until a temporal resolution of 1 Hz. Although in previous experiments we have demonstrated the operation of 20 UAS simultaneously with a flight permit in a specific category, in the present study only 5 UAS were operated in the open category of the EASA regulations.

### 2.2 Measurement site

The measurements were conducted at an Enercon E-82 WT with a rated power of 2 MW. The WT is located in a complex terrain on an elevated plateau. A prominent slope at 0.5 km to the west with an elevation of about 180 m, resulting from a river valley, dominates the topography (see Fig. 1). The wind direction at the site is dominated by westerly winds (see wind rose in



Fig. 1 extracted from the New European Wind Atlas, NEWA). Due to the complex terrain, the WT operates at a comparably

high hub height of 138 m with a rotor diameter of $D = 82$ m. The wind direction is measured by a sonic anemometer on the nacelle. Operating data from the WT are available throughout the SCADA system (Supervisory Control And Data Acquisition system), but not explicitly presented in this study due to confidentiality agreements.

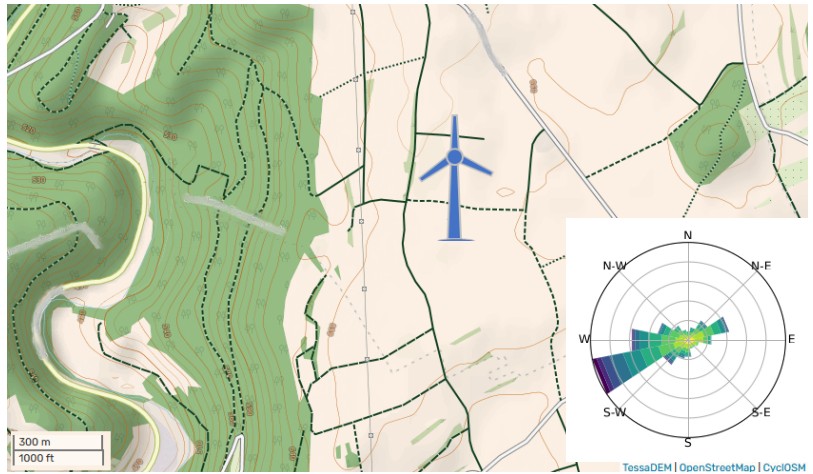

**Figure 1.** Topographical map of the measurement site including the location of the WT and the wind rose from the site. © OpenStreetMap contributors 2022. Distributed under the Open Data Commons Open Database License (ODbL) v1.0.

## 2.3    Flight strategy

In total, we carried out more than 80 UAS fleet flights on seven measurement days in 2022. We performed flights both in

the early morning under stable atmospheric conditions and under unstable conditions during the day. During the flights, wind speeds between 5 and 13 m s$^{-1}$ were observed from westerly directions in most cases.

   Different flight patterns were performed to study the wake of the WT and to measure the free flow at the same time. In the first pattern, referred to as the "longitudinal pattern", the UAS are horizontally distributed in streamwise (longitudinal) direction at an altitude of 120 m a.g.l. downstream of the WT. Note that this height is slightly lower than the hub height which

is at 138 m. Due to the operation in the open category, we were limited to flight altitudes of 120 m at far distances from the WT. Multiple UAS are positioned in this horizontal longitudinal line at different distances from the WT (up to $\Delta x = 3\,D$) with horizontal spacing between the UAS of $\Delta x = 0.5\,D$. This pattern is illustrated by the blue diamonds in Fig. 2. Additionally to the wake measurement, the inflow is measured at a longitudinal distance of $2\,D$ upstream of the WT (illustrated by the green diamond in the 'inflow' pattern in Fig. 2). On the UAS' way to the measurement height at 120 m a.g.l. the thermal stratification

is measured during the ascent. This means that a vertical profile of the WT inflow is also available at the beginning of each wake measurement. The goal of a second flight pattern in the wake is to measure the horizontal profiles of the wake. In this so-called "lateral pattern", multiple UAS are distributed laterally to the main wind direction in the wake of the WT. The lateral



positions of the UAS (relative to the WT nacelle) are chosen so that one UAS measures in the freestream at $\Delta y = 1 \, D$ and the remaining UAS measure inside the WT wake. The lateral spacing within the wake is designed to resolve the edge region

of the wake in particular. This pattern is conducted in different longitudinal distance to the WT from $\Delta x = 0.5$ to $1.5 \, D$. The orientation of the pattern was chosen based on the freestream wind direction measured by the UAS and the current orientation of the WT. Before each launch of the UAS-fleet, the orientation of the pattern was updated to ensure alignment to the current wind direction.

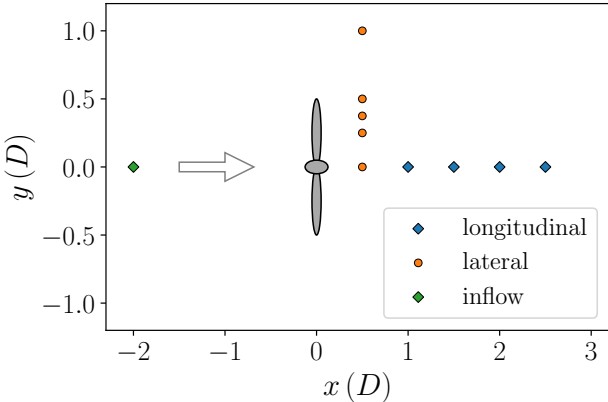

**Figure 2.** Different flight pattern of the UAS fleet from the top view. The arrow represents the wind direction towards the WT.

The wind direction and the orientation of the WT in atmospheric boundary layer flows are not stationary and can change

frequently. For this reason, misalignment in field measurements of an operational WT cannot be precluded. The definition of flight pattern alignment and WT misalignment are introduced in Fig. 3. $\gamma$ is defined as the yaw misalignment of the WT rotor to the incoming flow. Due to a yaw misalignment, the wake is deflected laterally by the deflection angle $\theta$. In addition to the characteristics of the inflow, the deflection angle depends on the thrust coefficient $C_T$ of the WT. A higher thrust coefficient leads to a higher deflection of the wake (Bastankhah and Porté-Agel, 2016). Additionally, the misalignment of the UAS flight

pattern with the incoming wind direction is defined as $\beta$.

## 3 Methods

### 3.1 Characterization of WT inflow

For characterization of the inflow and the atmospheric conditions, various parameters are determined and defined in the following. As mentioned in the introduction, the inflow significantly influences the characteristics of the WT-wake. From the inflow

pattern ($2 \, D$ upstream), the vertical profile is used for thermal stratification classification and a 10-minute averaging period at the final measurement position for wind and turbulence properties. First, the mean wind velocity $\bar{u}$ and mean wind direction $\bar{\Phi}$ are calculated for the 10 min hover time in the freestream. From the mean wind direction and the mean orientation angle of



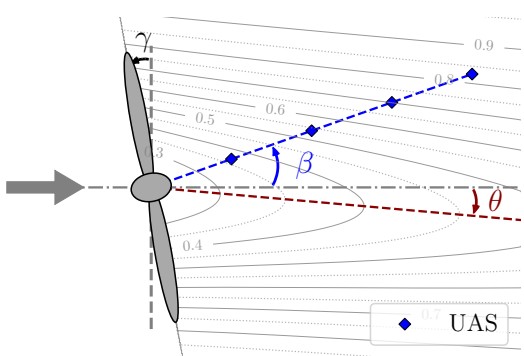

**Figure 3.** Misalignment of the WT and the UAS fleet from the top view. The reference inflow wind direction is represented by the arrow. $\gamma$ defines the yaw misalignment of the WT. The deviation of the orientation of the UAS pattern (blue diamonds) is defined by the angle $\beta$. The resulting wake deflection, due to the yaw misalignment, is defined by the deflecting angle $\theta$. The velocity deficit in the wake is illustrated by contour lines and extracted from FLORIS (NREL, 2021)

.

the WT, the actual alignment of the orientation of the pattern $\beta$ and the mean yaw misalignment of the WT $\gamma$ can be derived (see Fig. 3). Furthermore, the standard deviation of the inflow wind direction $\sigma_\Phi$ is listed as parameter in Table 1, since it is a

measure of the unsteadiness of the flow and influences the accuracy of the relative position of the UAS point measurements in the wake.

The turbulence intensity $I$ is used as the measure of the turbulence level. $I$ is defined as the standard deviation $\sigma_u$ normalized by the mean velocity $\bar{u}$

$$ I \quad = \quad \frac{\sigma_u}{\bar{u}} \quad . \tag{1} $$

In the following we refer to the streamwise turbulence intensity $I$ by taking only the streamwise velocity $u$ into account. Additionally, in Table 1 the variance $\sigma_u^2$ of the inflow is listed, since it accounts for a significant proportion of the turbulent kinetic energy. The thermal stratification is derived from the lapse rate $\Gamma$ in the corresponding heights of the WT (from 80 m to 120 m). The lapse rate is defined by the gradient of the virtual potential temperature $\Delta\theta_v$ with altitude $\Delta z$

$$ \Gamma \quad = \quad \frac{\Delta\theta_v}{\Delta z} \quad . \tag{2} $$

The atmospheric boundary layer (ABL) is divided into convective (CABL) for negative laps rate ($\Gamma < -0.2 \, \mathrm{K}/100 \, \mathrm{m}$), near-neutral (NABL) for close to zero ($|\Gamma| < 0.2$) and stable (SABL) for positive laps rate ($\Gamma > 0.2$).



**Table 1.** Flight protocol of considered flights in Sect. 4. The first number of the flight no. indicates the flight day (for instance flight 611 is carried out on flight day 6). The variables are calculated from the reference measurement and the definitions are listed below: $\gamma$ yaw-misalignment, $\beta$ deviation of pattern orientation, $\bar{u}$ mean velocity, $\Phi$ mean wind direction, $\sigma_\Phi$ standard deviation of wind direction, $I_x$ streamwise turbulent intensity, $\sigma_u^2$ streamwise velocity variance, atmospheric boundary layer (ABL) with convective (CABL), stable (SABL) and neutral (NABL).

| date | time | flight no. | $\gamma$ | $\beta$ | pattern | $\bar{u}$ | $\Phi$ | $\sigma_\Phi$ | $I_x$ | $\sigma_u^2$ | ABL |
|------|------|-----------|----------|---------|---------|-----------|--------|---------------|-------|--------------|-----|
| | [UTC] | | [°] | [°] | | [m s$^{-1}$] | [°] | [°] | [-] | [m$^2$ s$^{-2}$] | |
| 18.02.2022 | 11:05 | 206 | 7.3 | 7.1 | lateral | 11.57 | 242 | 11 | 0.214 | 5.85 | CABL |
| 18.02.2022 | 11:24 | 207 | 6.9 | 4.1 | lateral | 12.43 | 239 | 11 | 0.216 | 7.09 | CABL |
| 28.04.2022 | 10:30 | 407 | 21.5 | 9 | lateral | 6.92 | 79 | 11 | 0.166 | 1.53 | CABL |
| 28.04.2022 | 11:26 | 409 | 15.5 | -15.4 | lateral | 6.47 | 75 | 11 | 0.144 | 0.83 | CABL |
| 28.04.2022 | 11:47 | 410 | 21.6 | 3 | lateral | 6.42 | 73 | 11 | 0.157 | 1.2 | CABL |
| 12.05.2022 | 4:13 | 604 | 20.6 | 19.2 | longitudinal | 6.96 | 264 | 4 | 0.073 | 0.28 | SABL |
| 12.05.2022 | 4:34 | 605 | 16.2 | 18.2 | lateral | 6.5 | 264 | 5 | 0.074 | 0.28 | SABL |
| 12.05.2022 | 4:53 | 606 | 7.6 | 6.4 | lateral | 6.18 | 257 | 8 | 0.089 | 0.35 | SABL |
| 12.05.2022 | 5:14 | 607 | 15.7 | 8.9 | lateral | 6.08 | 259 | 8 | 0.091 | 0.34 | SABL |
| 12.05.2022 | 5:35 | 609 | 20 | 16.3 | longitudinal | 6.04 | 261 | 8 | 0.111 | 0.49 | NABL |
| 12.05.2022 | 6:00 | 611 | 18.5 | 12.8 | longitudinal | 7.1 | 263 | 8 | 0.112 | 0.62 | NABL |
| 12.05.2022 | 6:20 | 613 | 14.1 | 7.8 | longitudinal | 7.05 | 258 | 9 | 0.115 | 0.96 | NABL |
| 07.11.2022 | 7:10 | 702 | 13 | 4.9 | lateral | 9.04 | 245 | 7 | 0.119 | 1.14 | NABL |
| 07.11.2022 | 7:31 | 703 | 13.9 | 0.6 | lateral | 9.17 | 240 | 7 | 0.132 | 1.43 | NABL |
| 07.11.2022 | 7:53 | 704 | 17.7 | 2.6 | lateral | 9.29 | 242 | 6 | 0.124 | 1.55 | NABL |
| 07.11.2022 | 8:20 | 706 | 19.5 | 5.8 | longitudinal | 7.48 | 246 | 7 | 0.172 | 1.7 | NABL |
| 07.11.2022 | 9:06 | 708 | 15.2 | 6.4 | longitudinal | 6.97 | 246 | 8 | 0.207 | 2.15 | NABL |
| 07.11.2022 | 9:25 | 710 | 16.2 | 12.4 | longitudinal | 7.77 | 252 | 8 | 0.142 | 1.19 | NABL |
| 07.11.2022 | 9:50 | 711 | 8.9 | 4.6 | lateral | 8.66 | 245 | 8 | 0.144 | 1.56 | NABL |
| 07.11.2022 | 10:12 | 712 | 10.9 | 5.2 | lateral | 7.76 | 245 | 8 | 0.124 | 1.11 | NABL |
| 07.11.2022 | 10:39 | 713 | 13.1 | 8.4 | lateral | 7.74 | 249 | 9 | 0.151 | 1.44 | NABL |

## 3.2 WT wake analysis

In order to analyze the WT wake, more parameters need to be defined in this section. One important parameter for classifying the WT operation point is the tip speed ratio $\lambda$. The tip speed ratio (TSR) is defined by the tip speed calculated from the rotor

diameter $D$ and the angular velocity $\omega$ ($\omega = \Omega/2\pi$) divided by the freestream velocity $u_0$

$$\lambda = \frac{\omega\, D/2}{u_0} \quad . \tag{3}$$





The frequency of occurrence of the tip vortex at a fixed position is called the blade passing frequency $f_{\mathrm{bp}}$, which is defined by the rotational speed $\Omega$ of the rotor and the number of blades $n_b = 3$.

$$f_{\mathrm{bp}} = \Omega\, n_b \quad . \tag{4}$$

If the rotational speed is present in rpm (revolutions per minute) it needs to be transformed to revolutions per second. The turbulence intensity which is added to the freestream turbulence by the WT is called *added turbulence intensity* $\Delta I$. It is calculated from the freestream turbulence $I_0$ and the measured turbulence intensity inside the wake $I$ (Frandsen, 2007)

$$\Delta I = \sqrt{I^2 - I_0^2} \quad . \tag{5}$$

The horizontal momentum flux $\overline{u'v'}$ is defined by the covariance of the horizontal wind components $u$ and $v$ divided by the

number of time steps $N$

$$\overline{u'v'} = \frac{\sum (u_i - \bar{u})(v_i - \bar{v})}{N} \quad . \tag{6}$$

An analytical estimation of the near wake length $l_{nw}$ is conducted in order to distinguish whether we are measuring in the near or far wake. The equation is derived from wind tunnel experiments by Bastankhah and Porté-Agel (2016)

$$\frac{l_{nw}}{D} = \frac{\cos\gamma\,(1 + \sqrt{1 - C_T})}{\sqrt{2}(c_\alpha\,I + c_\beta\,(1 - \sqrt{1 - C_T}))} \quad , \tag{7}$$

with the constant parameters $c_\beta = 0.154$ and $c_\alpha = 3.6$ (Fuertes et al., 2018). From this equation for an unstable condition with high turbulence intensity (# 206) the length is $l_{nw} = 1.7\,D$ while for a stable case (# 604) the near wake length is $l_{nw} = 2.9\,D$.

In order to estimate the influence of the yaw misalignment on the wake deflection, an analytical dependency is presented. This estimation is based on the conservation of momentum and mass and is a function of the downstream distance $x$ and the yaw misalignment $\gamma$ (Jiménez et al., 2009).

$$\theta = \frac{\sin\gamma\,C_T}{2\,(1 + \zeta \frac{x}{D})^2} \quad , \tag{8}$$

where $\zeta$ is the wake growth rate. Jiménez et al. (2009) defined $\zeta = 0.1$ for yaw misalignments smaller than $\gamma = 20°$. For the considered flights the mean maximal yaw-misalignment is about $\gamma = 20°$. Together with an approximate $C_T = 0.7$ the deflection is $\theta = 5.7°$ at a downstream distance of $1\,D$.

## 4   Results

At the field site we performed multiple flight strategies in different atmospheric conditions over several days. Two flight strategies are considered in detail, namely the longitudinal and lateral pattern. In this section, we first give an overview of all performed flights. This is followed by a detailed analysis of a single flight considering the time series of a lateral pattern and the associated turbulence spectra. The lateral profile of the velocity, the turbulence intensity and the horizontal fluxes for stable to near-neutral atmospheric conditions are examined in the middle section. The downstream evolution of the wake is

studied using the longitudinal flight pattern. Finally, the results of the lateral pattern under unstable atmospheric conditions are compared to stable conditions.





## 4.1 Overview flight data

In Fig. 4 all considered individual UAS flights are shown by a single point at their horizontal measurement position. The coordinate system of UAS locations is aligned with the reference wind direction and the WT. For example, if the alignment
of the longitudinal pattern does not match the reference wind direction measurement, this discrepancy is visible through an angle of the UAS pattern alignment compared to the longitudinal centerline of the wake. The normalized wind velocity is indicated by the marker color. Depending on the pattern, the wind velocity is normalized either with UAS measurement upstream ($\Delta x = 2\ D$) for the longitudinal pattern or with measurements in the freestream ($\Delta y = 1\ D$) for the lateral pattern. Thus, the normalized velocity shows the velocity deficit measurements in the wake. This figure clearly shows qualitatively the
wind deficit in the WT-wake at different positions. The velocity deficit in longitudinal and lateral direction is examined in more detail in the following sections. The obvious trend in the misalignment between the pattern orientation and the freestream wind direction $\beta$ is due to the trade-off between wind direction and WT orientation. The different angles are illustrated in Fig. 3 and reasons for the differences are discussed in the following section.

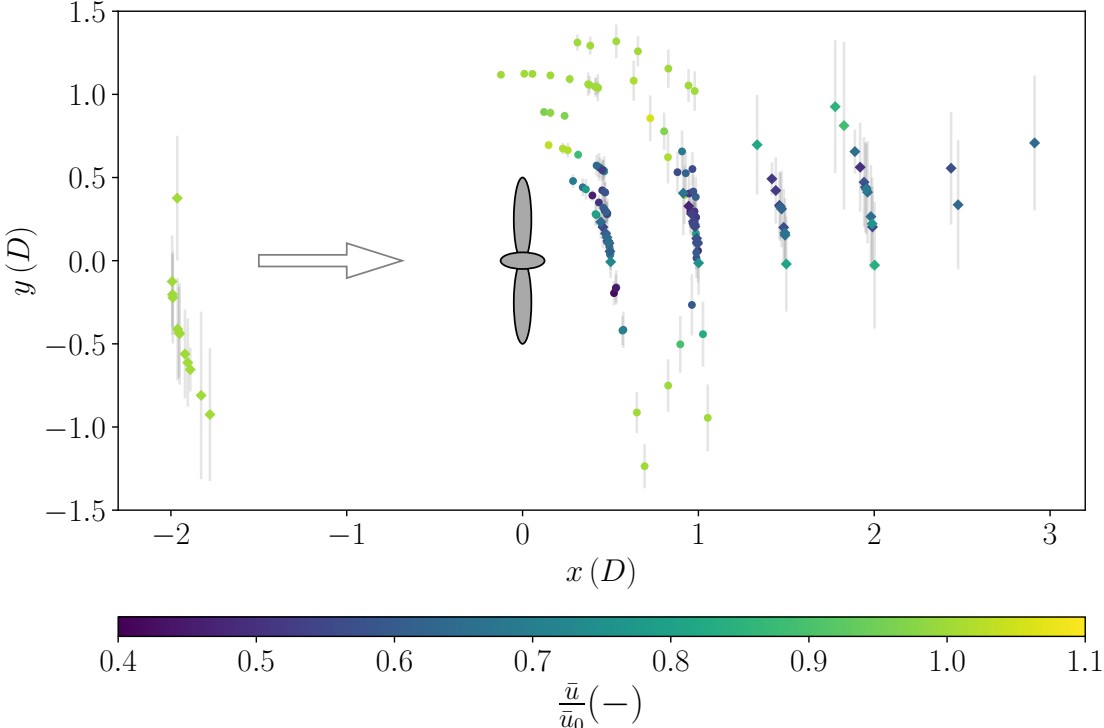

**Figure 4.** Normalized wind velocity of UAS measurements $\bar{u}/\bar{u}_0$ at locations relative to the reference wind direction. The black bar represents the position and dimensions of the WT rotor. Diamonds marker represent the longitudinal flight pattern while circle markers define the lateral flight pattern. The arrow indicates the inflow wind direction. The standard deviation in lateral position is illustrated by the grey vertical lines and is calculated from the standard deviation of the inflow wind direction.





## 4.2 Reference measurements

On the measurement site only a single WT exist without any further measurement devices, such as meteorological mast or lidars. In order to study the wake of a WT, it is essential to measure the ambient conditions during the wake measurements. Therefore, additionally to each wake measurement, a reference measurement is conducted with the UAS-fleet. As already mentioned, the reference is measured either two rotor diameters upstream in the inflow or one diameter in lateral distance to the side of the WT. After a slow ascent, during which a vertical profile of all thermodynamic variables can be measured, the

UAS hovers for 10 minutes at the top altitude to determine a reference wind direction and wind speed. In Fig. 5 the measured reference wind direction is compared with data from the WT SCADA system. The wind direction measurements of the WT and the UAS are well correlated (correlation coefficient of $R = 0.98$). It is obvious that a systematic bias between the independent measurements exists. Possible reasons are:

1. Spatial distance between the measurements.

2. Imperfect calibration of flow distortions for the sonic on the WT.

3. Errors in northing of the either UAS or WT sonic.

4. Temporal offset between measurements, since WT-data is only available as 10-minute average and does not always perfectly align with the UAS flights.

Possibility 1 and 4 would rather manifest in a random error, while possibility 2 and 3 can yield a systematic error. The calibra-

tion and orientation of the WT sonic is not known to the authors so that it can only be guessed that a combination of all reasons causes the bias and scatter between UAS and WT sonic. Comparing the yaw angles of the WT in Fig. 5 with the UAS measurements, a mean offset throughout all flights can be observed as well. This indicates that the yaw controller is not perfectly adjusted for this specific WT. The trend in the misalignment from the WT leads to the relative position deviations in the Fig. 4, since it was attempted to align the UAS pattern with both, the turbine orientation and the wind direction. It is worth noting that

no systematic error between UAS measurements in the freestream upstream and lateral is observed. Reference measurements with the UAS fleet did also not show systematic errors of any UAS beyond the uncertainties that were previously observed for the system (Wetz and Wildmann, 2022).

## 4.3 Wake measurements under stable to near-neutral atmospheric conditions

In the following subsection we focus on the lateral and longitudinal flight pattern under stable to near-neutral atmospheric

conditions. Details about the considered flights in this section are listed in Table 1.

### 4.3.1 Analyses of a single lateral flight pattern

The time series of flight # 606 is shown as an example in Fig. 6. The flight was conducted in the early morning (04:53 UTC) of 12 May 2022, before the nighttime stable ABL was completely eroded by turbulence of the growing mixed layer. As introduced





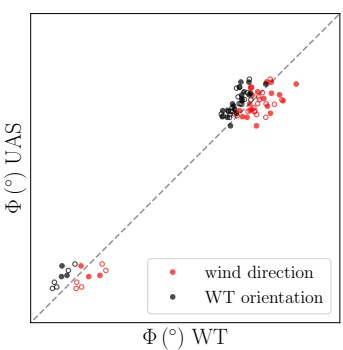

**Figure 5.** Comparison of WT wind direction and orientation measurements with UAS reference measurements. UAS measurements are conducted in the freestream, either lateral $y = 1\,D$ (empty dots) or upstream $x = 2\,D$ (filled dots) of the WT. Both, the wind direction and the nacelle orientation are measured on the WT nacelle.

in Sect. 2, the lateral pattern was arranged so that one UAS is located in the freestream, one is placed at the edge of the wake, and the remaining are laterally distributed in the WT-wake. In Fig. 6a the time series of horizontal velocity clearly show that the outer UAS ($\Delta y = 1.05$) measures in the freestream, indicated by the highest average velocity, while the inner UAS are placed inside the wake. The innermost UAS ($\Delta y = 0.06\,D$) behind the nacelle has a considerably lower wind speed deficit than those hovering between $y = 0.2\,D$ and $y = 0.45\,D$. This increase in velocity at the center of the wake is already indicative of a double Gaussian shape of the lateral wake profile. The measurement with the highest velocity fluctuations is located at the edge of the WT wake between $y = 0.5\,D$ and $0.6\,D$. In Fig. 6b, the wind direction of the reference UAS shows a strong variation of the inflow ($\sigma_\Phi = 8°$). This variation of inflow wind direction leads to a variation of the relative lateral measurement position, with respect to the reference wind direction. These variations of the lateral measurement positions are shown in Fig. 6c. As the velocity series of the inner UAS ($\Delta y = 0.3\,D$ and $\Delta y = 0.42\,D$) already suggests, the relative measuring positions are inside the wake over the entire flight. More interesting is the relative lateral position of the UAS at the edge of the wake. A correlation between the relative lateral position and the velocity deficit can be observed here (with correlation coefficient of $R = 0.5$). For example, taking the measurement at 04:59 UTC, both the lateral position $y > 0.6\,D$ and the velocity imply measurements outside the wake. On the other hand, after the wind direction change at around 05:00 UTC, both the lateral position and the measured velocity indicate measurements within the wake of the WT. This meandering of the wake evidently causes a high turbulence intensity measurement at the edge of the wake. These results show the sensitivity of the relative position of wake measurements in field experiments even and especially in the near wake region.

In order to understand the distribution of the turbulence energy across the scales, the power spectrum $S_u$ of streamwise wind velocity for the same flight (# 606) is shown in Fig. 7. Apart from the larger scales ($f < 0.03\,\text{Hz}$, $l > 200\,\text{m}$), the measurements inside the wake show in general a higher level of energy compared to the freestream (dashed black line), particularly in the small scales ($f > 0.3\,\text{Hz}$, $l < 20\,\text{m}$). We can assume that the added turbulence in the wake is the main reason for this increase.

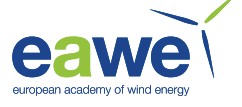


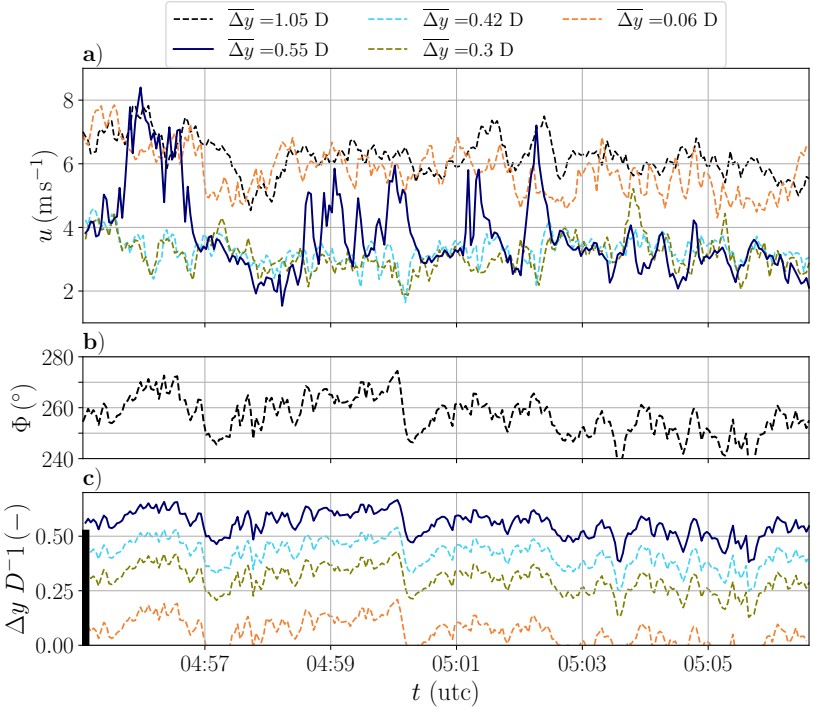

**Figure 6.** Time-series of wind velocity a), wind direction in b) and lateral position c) of a lateral flight pattern in stable conditions (fl. # 606). The lateral positions are calculated using the wind direction of reference UAS measurement, which is shown in the middle figure. The black bar in c) indicates the lateral position of the WT.

The measurement at the edge of the wake ($\overline{\Delta y} = 0.55$) show the highest turbulence level, especially at the larger scales, but also at smaller scales. This additional variance is caused by the wake meandering, which itself is mainly caused by the variation of the inflow wind direction. Another feature that can theoretically be observed at the edge of the wake is the tip vortex. In order to assess whether the tip vortex in the spectrum (Fig. 7) of the UAS measuring at the edge of the wake can be identified, we take the WT rotational speed into account. During the period of the considered flight, the mean rotational speed of the WT is 12 rpm,

resulting in a blade-passing frequency (BPF, also known as tip vortex shedding frequency) of $f_{bp} = 0.6$ Hz. In the area of the BPF in Fig. 7 an increases in the spectrum of the UAS at the edge of the wake ($\overline{\Delta y} = 0.55$) can be observed. Together with the freestream velocity of $\bar{u}_0 = 6.18$, the tip speed ratio results in $\lambda = 8.36$. Considering the advection velocity at this position of $u_{adv} = 3.6$ m s$^{-1}$, the axial spacing of the helical vortices is about 6 m ($0.07\,D$). In this case we use the mean velocity at the measurement position as advection velocity ($u_{adv} = \bar{u}$), while in wind tunnel experiments the instantaneous velocity

is often used (Zhang et al., 2011; Sherry et al., 2013) resulting in a higher advection velocity compared to the freestream velocity. Taking their ratios of $u_{adv}/u_0 = 0.8$ into account the vortex spacing would be slightly larger. Porté-Agel et al. (2019) specify a typical range of the normalized mean advection velocity of the tip vortices of $u_{adv}/u_0 = 0.73\ldots0.78$. However,





in our case, assuming a helical vortex spacing of 6 m, at a longitudinal distance of $x = 0.5\ D$, where the measurements are taken, approximately seven tip vortices could be observed between the measurement position and the WT. Thus, the measured

tip-vortex has a rotation trajectory history of more than two complete WT revolutions. Given such a trajectory history, the peak in the turbulence spectrum is not expected to be very pronounced in the present case. In addition, the rotational speed of the WT and the wind speed in field operation are not constant, as it is mostly the case in wind tunnel experiments, which can also cause some blur in the spectrum.

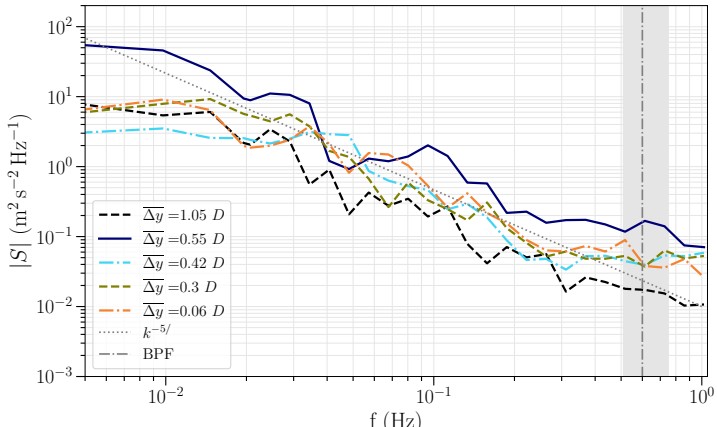

**Figure 7.** Power spectra of streamwise velocity $S_u$ for flight # 606 (lateral pattern) at different lateral positions $\Delta y$. The blade-passing frequency (BPF) is indicated by the vertical dashes dotted line. The grey background around the BPF represents the bandwidth of the BPF based on the extremes of WT revolution speed. The spectra are processed with bin-averages in the frequency-space to decrease the noise.

### 4.3.2 Horizontal wake profile of velocity deficit and turbulence intensity

For the lateral pattern, the wind velocity deficit is calculated using the reference UAS, which is located at a lateral distance of $\Delta y = 1\ D$ to the WT. The pattern is conducted at different longitudinal distances to the WT and in slightly different lateral arrangements. The results in Fig. 8a show the normalized velocity of multiple flights of the lateral pattern on 12 May 2022 under stable and on 07 November 2022 under near-neutral thermal stratification as a function of the lateral distance to the WT hub. The lateral distances differ slightly for identical flight patterns, since we take the misalignment $\beta$ of the pattern to the

wind direction into account. The measuring points of the individual flights, consisting of maximum five UAS, are connected in Fig. 8.

The shape of the horizontal velocity profile in a downstream distance of $\Delta x = 0.5\ D$ follows an approximate double Gaussian shape, assuming a nearly symmetrical profile for the remaining parts on the opposite lateral side (see Fig. 8a). A double Gaussian velocity distribution is characterized by a strong decrease in velocity at the edge of the wake between $\Delta y = 0.3\ D$

and $0.6\ D$ followed by an increase in velocity (up to almost free-stream velocity) in the center of the wake. This shape is





closely connected to the energy that the WT extracts from the wind. The majority of the energy is typically converted at the outer areas in spanwise direction of the rotor, resulting in a comparatively low velocity deficit in the center in the near wake region. Further downstream, at $x = 1\ D$, the shape differs for the considered flights. The flights on day 7 (# 7xx) show a less pronounced double Gaussian shape. This could be explained by the different freestream turbulent intensities and the thermal

stratification. On day 6, the turbulence intensity was lower than on day 7, so that vortices remain stable for longer distances. At $x = 1\ D$, we still see the same principle shape and relationship between the two days, only small changes in magnitude of the velocity deficit can be observed compared to $x = 0.5\ D$.

Figure 8b shows the normalized turbulence intensity $I/I_0$ over the lateral distance for the same flights. The highest turbu-

lence intensity can clearly be observed at the edge of the wake around $\Delta y = 0.5\ D$ due to the tip vortices of the WT blades, the shear layer and the meandering of the wake. At a spatially fixed measurement position at the edge of the WT, the meandering of the WT-wake causes high variation of wind velocity due to temporal change in the relative position from inside to outside of the wake (see Sect. 4.3.1). However, towards the wake center $I$ decreases, showing a double Gaussian shape similar to the velocity deficit, which has also been reported in literature (Maeda et al., 2011). On day 6 the freestream turbulence intensity

was lower than on day 7. The increase of normalized turbulence intensity can therefore be higher than on day 6, which is observed at $0.5\ D$ and $1\ D$. The peak of turbulence intensity further downstream at a distance of $x = 1\ D$ is less pronounced compared to $x = 0.5\ D$.

### 4.3.3 Streamwise development of the wake

In Fig. 9a the development of the normalized velocity and in Fig. 9b the normalized turbulence intensity in streamwise direction

of the longitudinal pattern is shown. The measurements are normalized with the inflow measurement at an axial distance of $\Delta x = 2\ D$ upstream of the WT. Due to the mentioned pronounced double Gaussian profile of the lateral velocity distribution at $x = 0.5\ D$, the velocity in the center of the wake is comparatively high. As the velocity deficit profile turns into a single Gaussian shape further downstream (from $0.5\ D$ to $1\ D$) due to turbulent mixing, the velocity at the wake center decreases. Even further downstream, the accuracy of the pattern orientation in our experimental setup plays a major role. For example, a

pattern misalignment of $\beta = 10°$ indicates a lateral displacement error of $\Delta y = 0.35\ D$ in a longitudinal distance of $x = 2\ D$. This displacement leads to measurements towards the lateral edge of the wake, where the velocity increases compared to the center of the wake, assuming a single Gaussian velocity distribution in these downstream distances. Both effects are most evident for flight # 604: the turbulence intensity ($I_0 = 0.073$) and the standard deviation of the wind direction ($\sigma_\Phi = 4°$) inflow are low compare to the other considered flights; and the misalignment of the pattern is comparable high ($\beta = 19°$). The

low level of turbulence intensity and wind direction variation leads to a pronounced double Gaussian distribution, resulting in a prominent drop of velocity from $x = 0.5\ D$ to $x = 1\ D$ in the wake center. Further downstream, the large misalignment causes a relative measurement location almost outside the wake, which leads to the relatively large increase in velocity of flight # 604.





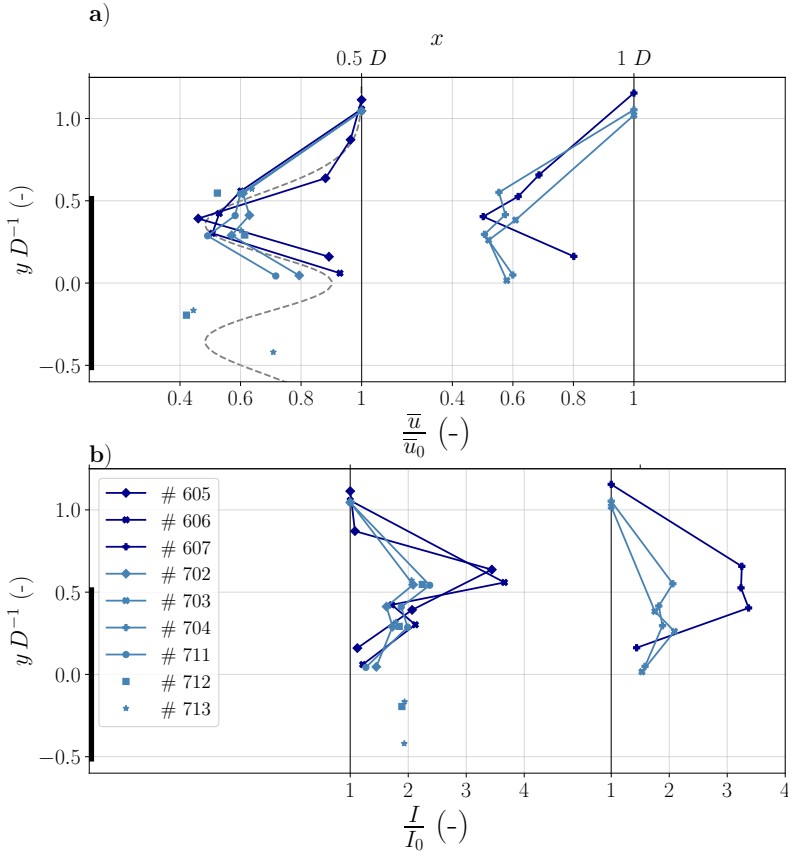

**Figure 8.** a) Lateral profile of normalized wind speed $\bar{u}/\bar{u}_0$ in the WT wake at different downstream distances ($x = 0.5\ D$ and $1\ D$). b) Lateral profile of normalized turbulence intensity $I/I_0$ at the bottom figure. Different colors indicating different flight days with both stable (dark blue) to near-neutral conditions (light blue). Details about the considered flights are found by the flight number in table 1. The grey dashed line indicating a mean double Gaussian symmetric fit of the present data.

Following the same argumentation, the extreme behaviour of flight # 604 of the normalized turbulence intensity (Fig. 9b)

can be explained by the misalignment of the pattern $\beta$ and the low freestream turbulence intensity $I_0$. In general, the turbulence intensity increases with downstream distance in the near wake center. This can be explained by the turbulent mixing of the high turbulence region on the edge of the wake towards the center. For this experiment, the deviation in pattern orientation causes the measurement to be taken outside of the wake center, further towards the edge of the wake, which typically exhibits higher turbulence intensities. Flight # 708 shows a slight decrease in turbulence intensity over longitudinal distance, which

can be explained by the high level of turbulence intensity in the free stream ($I_0 = 0.207$). Due to the already high turbulence intensity in the ambient flow, the turbulence induced by the WT plays a minor role in the absolute value of turbulence intensity in the wake. The downstream position of maximum turbulence intensity in the wake center typically occurs at the transition from near-wake to far-wake. Since the extent of the near wake is larger at low freestream turbulence intensities, the maximum

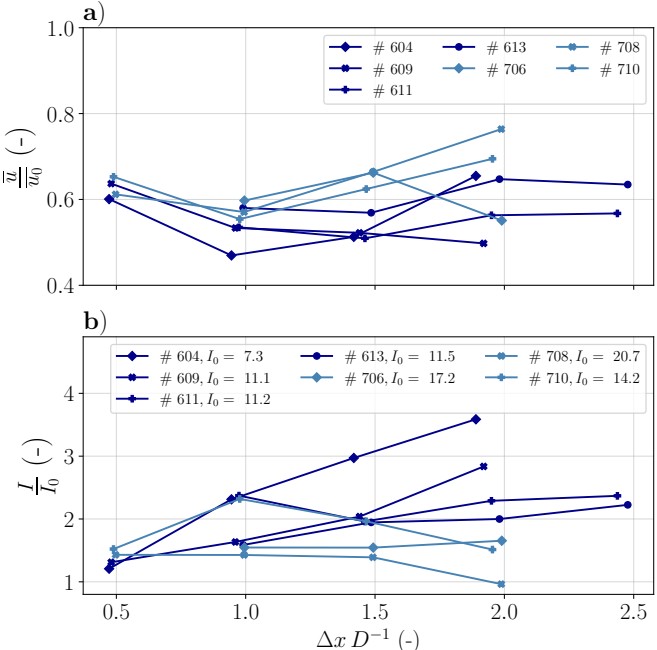

**Figure 9.** a) Normalized wind speed $\bar{u}/\bar{u}_0$ and b) normalized turbulence intensity $I/I_0$ is shown in the center of WT wake at different downstream distances. Different colors indicating different flight days with both stable (dark blue) to near-neutral conditions (light blue). The reference turbulent intensity $I_0$ given in the legend is shown in %.

of turbulence intensity will arise further downstream compared to high ambient turbulent intensities (Wu and Porté-Agel, 2012).


Due to the counter clockwise rotating velocity field (viewed from upstream) in the near wake, a measurable lateral velocity component is expected below the center of the wake, where measurements are taken in the longitudinal pattern. Since the lateral velocity is defined positive towards north, the lateral velocity at this position is expected to be negative (for westerly winds). This negative lateral velocity close behind the turbine at a longitudinal distance of $\Delta x = 0.5\ D$ and 18 m ($\Delta z = -0.2\ D$)

below the center of the wake is observable in Fig. 10 for all cases. Furthermore, the development of the lateral velocity over the longitudinal distance in the wake is shown there. Due to the sensitivity of the exact measurement position to the lateral velocity component, a detailed interpretation is not given at this point. Overall, the velocity field perpendicular to the mean flow is expected to decrease further downstream. In particular at the lower part of the wake, due to strong turbulent mixing, a decrease of the wake rotation is assumed. Zhang et al. (2011) showed in wind tunnel tests that the lateral velocity decreases

significantly from 1 D towards 2 D downstream distances. At $\Delta x = 5\ D$ the wake rotation is no longer observable.

The analyses of the longitudinal flight pattern show the difficulty to take in-situ measurements of the far-wake in a field experiment at a complex site, even with a flexible measurement system like the SWUF-3D fleet.

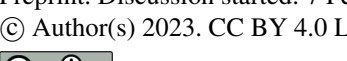



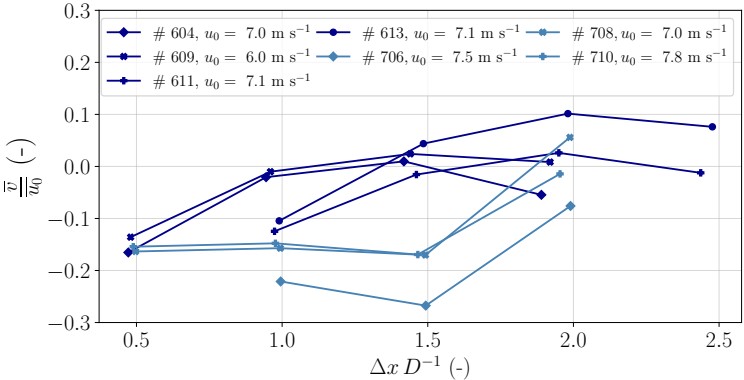

**Figure 10.** Normalized lateral wind speed $\bar{v}/\bar{u}_0$ in the center of WT wake at different downstream distances. Different colors indicating different flight days with both stable (dark blue) to near-neutral (light blue) conditions.

#### 4.3.4 Horizontal momentum fluxes of lateral distributed measurements

In wind energy science, predicting WT wake decay is a key feature for predicting wind farm efficiency. Turbulent fluxes

are a major process that drives the energy transport from the free flow towards the wake of the WT. The turbulent fluxes therefore have a direct impact on the wake recovery. In the following we examine the horizontal momentum fluxes $\overline{u'v'}$ for laterally distributed measurements in two different downstream distances ($x = 0.5\,D$ and $x = 1\,D$), again only for stable to near-neutral atmospheric conditions. In Fig. 11 the horizontal fluxes are normalized by the square of the freestream velocity $\bar{u}_0^2$ comparable to Bastankhah and Porté-Agel (2017). At the edge of the wake at $y = 0.6$ negative fluxes are observed, which

lead to an entrainment of energy into the wake. On the other side of the wake, the sign of the fluxes are positive for the same reason. Closer to the center of the wake, the signs of the fluxes are in the opposite direction due to energy transport from the center with high wind speed towards the low speed area at the edge of the wake. In $1\,D$ distance the momentum fluxes from the inner wake towards the outer wake are less prominent or no longer observable. In the stable case a high momentum flux towards the wake is still present at the edge of the wake.

### 4.4 Wake measurements in unstable atmospheric conditions

The studied results so far were only based on stable to near-neutral atmospheric conditions where wakes are known to be most critical for wind farm operation. However, unstable, convective conditions occur frequently, particularly at onshore sites in complex terrain. Therefore, in this section we will additionally look at some flights in unstable conditions. In Fig. 12 the unstable flights are added to the previously shown lateral profile of the normalized velocity and the normalized turbulence intensity.

It is evident that for unstable conditions the double Gaussian shape at $x = 0.5\,D$ and $x = 1\,D$ is no longer recognizable for both the velocity and turbulence intensity distribution. The standard deviation of the incoming wind direction is higher in the

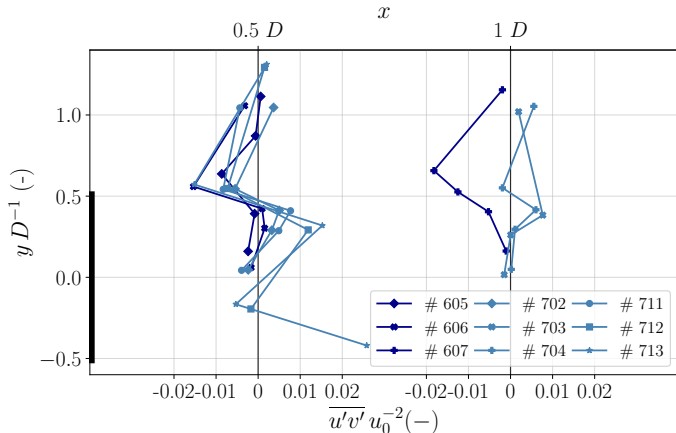

**Figure 11.** Lateral profile of normalized horizontal momentum fluxes $\overline{u'v'}/\bar{u}_0^2$ in the WT wake at different downstream distances ($x = 0.5\ D$ and $1\ D$). Different colors indicating different flight days with both stable (dark blue) to near-neutral conditions (light blue).

unstable cases (approx. $\sigma_\Phi = 11$) than in the stable cases ($\sigma_\Phi = 6 \ldots 8$). In addition, the higher turbulence in the ABL leads to a more pronounced turbulence mixing in the near wake and a rapid breakup of tip vortices. All mentioned effects smooth the lateral profile towards a single Gaussian shape. A direct comparison of the normalized turbulence intensity profile between

stable and unstable cases is difficult since the freestream turbulence intensity $I_0$ is much higher in the unstable cases. Therefore, the added turbulence intensity $\Delta I$ (see Eq. 5) is shown in Fig. 13. The added turbulence intensity supposedly isolates the wake induced turbulence intensity from the freestream turbulence, making different atmospheric conditions more comparable. The comparison reveals that the amount of added turbulence intensity is about the same in stable and unstable cases. However, in the center region of the wake $y < 0.3\ D$ the added turbulence intensity decreases more significantly under stable conditions,

which is a consequence of the double-Gaussian wake shape as presented before. Since the double Gaussian shape is not clearly observable under unstable conditions in the measured distances, it is expected that weaker horizontal fluxes can be observed from the wake center towards the edge of the wake. In Fig. 14, flights under unstable atmospheric conditions are compared with stable cases. Comparable to the stable conditions, the fluxes towards the wake are observable at the edge of the wake ($y = \pm 0.5\ D$) in CABL. However, the opposite direction of horizontal fluxes towards the edge of the wake in the center region

($y = \pm 0.3\ D$) are less pronounced. Overall the magnitude of the fluxes slightly increases in unstable conditions, due to stronger turbulent mixing.

## 5   Discussion

### 5.1   Horizontal velocity deficit distribution

In the present study we observed a double Gaussian horizontal distribution of the velocity deficit in the near wake under stable

atmospheric conditions. In the literature mostly the velocity distributions for distances greater than $1\ D$ are examined, only a



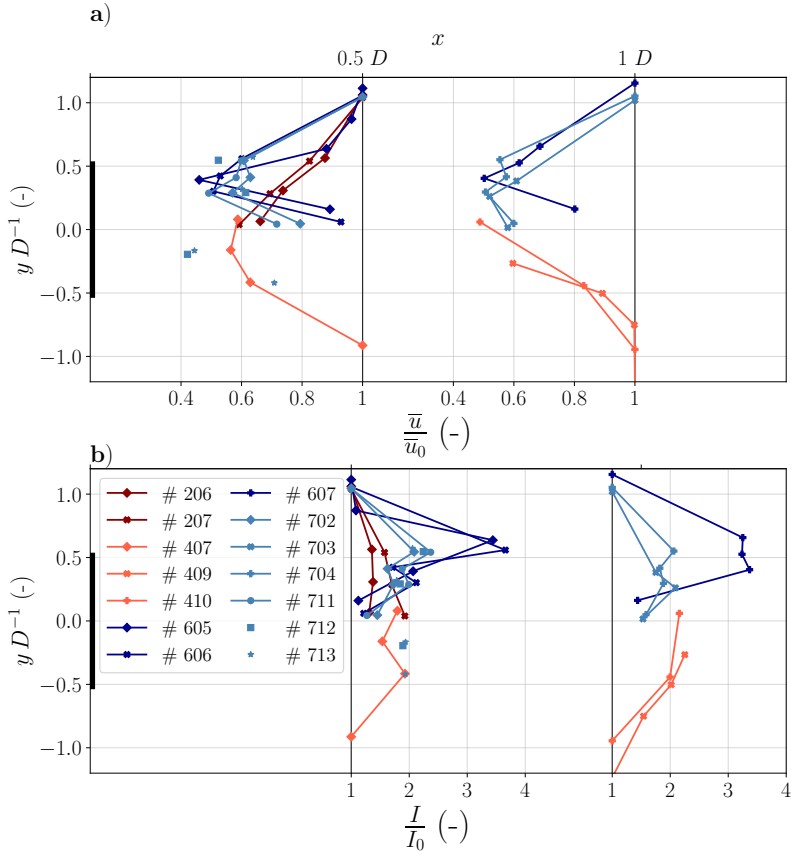

**Figure 12.** a) Lateral profile of normalized wind speed $\bar{u}/\bar{u}_0$ in the WT wake at different downstream distances ($x = 0.5\,D$ and $1\,D$). b) Lateral profile of normalized turbulence intensity $I/I_0$ at the bottom figure. Different colors indicating different flight days. The blue color represents stable to near-neutral atmospheric conditions while the red color indicates unstable convective conditions (dark red for highly unstable and light red for less unstable cases).

few publications exist for measurements close to the rotor plane. Menke et al. (2018) observed a double Gaussian distribution at $1\,D$ even in highly complex terrain using a ground-based wind lidar at the Perdigão site in 2015. Nacelle-based lidars are also often used for example by Herges and Keyantuo (2019) for downstream distances up to $\Delta x = 1\,D$, but generally they are mostly conducted for flow field studies further downstream $\Delta x > 2\,D$. Therefore, field data are rarely available especially in
the near wake region as close as $0.5\,D$ downstream.

Krogstad and Adaramola (2011) examined velocity profiles at the near wake in wind tunnel experiments with uniform inflows. They found that the velocity profile is strongly dependent on the tip speed ratio (TSR). For high tip speed ratios of about $\lambda = 8 \ldots 9$, a double Gaussian distribution was clearly observed, even with a slightly accelerated region in the center part of the wake. However, if the turbine is operated closer to the design tip speed ratio of $\lambda = 6$, the velocity profile becomes more
uniform and could be assumed to be a single Gaussian distribution. Transferring these results to our study, where the stable

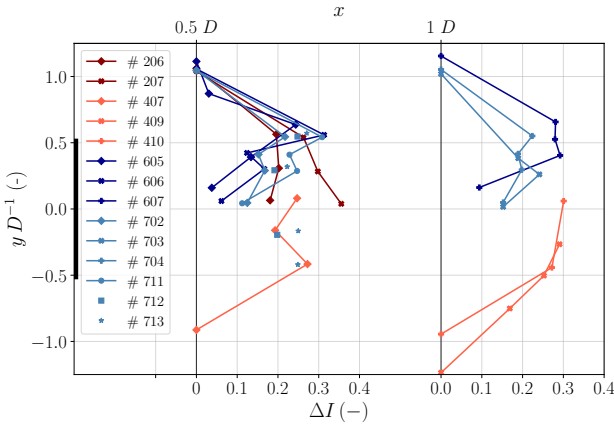

**Figure 13.** Lateral profile of added turbulence intensity $\Delta I$ in the WT wake at different downstream distances ($x = 0.5\,D$ and $1\,D$). Different colors indicating different flight days. The blue days representing stable to near-neutral atmospheric conditions while the red days indicating unstable convective conditions.

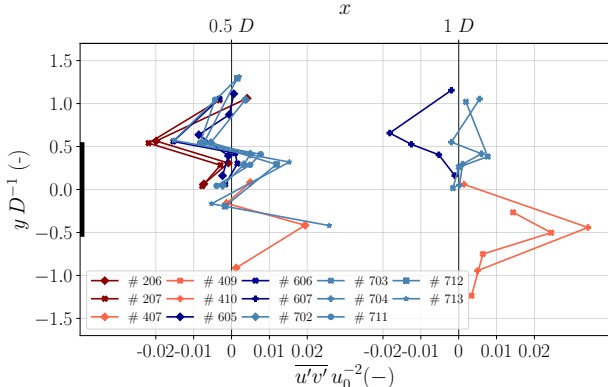

**Figure 14.** Lateral profile of normalized horizontal fluxes $\overline{u'v'}/\bar{u}_0^2$ in the WT wake at different downstream distances ($x = 0.5\,D$ and $1\,D$). Different colors indicate different flight days. Blue represents stable to near-neutral atmospheric conditions while red indicates unstable convective conditions.

and near-neutral cases were conducted at TSRs $\lambda = 7.5 \dots 9$, the high TSR could also drive the double Gaussian distribution. Under unstable conditions, the double Gaussian distribution is not observed in the present study (see Fig. 12). For the flights on day 2 under unstable conditions, the tip speed ratio was closer to the design point at $\lambda = 6$, which could explain the single Gaussian distribution according to Krogstad and Adaramola (2011). However, the other unstable measurements on day 4 were carried out at TSR of around $\lambda = 8$ which are comparable to the stable flights. Therefore, the moderate TSR cannot be the only explanation for the single Gaussian velocity distribution on day 4, but it is rather the unstable atmospheric condition of the inflow which influences the velocity profile in this case. Machefaux et al. (2015) examined the effect of atmospheric stability




on WT wakes using a nacelle-based pulsed lidar and LES. They clearly observed a dependency of atmospheric stability on the velocity deficit distribution. Under stable conditions, the velocity at the wake center increases significantly at a longitudinal

distance of $\Delta x = 1\,D$, which causes a double-Gaussian distribution. Whereas in unstable cases the velocity profile is almost flat with only a slight increase towards the wake center.

### 5.2 Horizontal fluxes

We have clearly observed lateral turbulent momentum fluxes towards the wake at the edge of the wake (see Fig. 14). These momentum fluxes are in agreement with Bastankhah and Porté-Agel (2017) who performed PIV measurements in wind tunnel

experiments for different tip speed ratios. The values for the normalized momentum fluxes $\overline{u'v'}/\bar{u}_0^2$ are of the same magnitude as in the present study. However, the turbulent fluxes from the center wake towards the edge of the wake, which can be seen in Fig. 14, are not observed in their wind tunnel experiments. The PIV measurements were limited to an approximate downstream distance of $x = 0.7\,D$, thus no results are discussed at $x = 0.5\,D$. Furthermore, the velocity distribution at $x = 1\,D$ in their study shows only a weak double Gaussian distribution, meaning that the lower velocity deficit in the center region is

less prominent. Therefore, the velocity gradients within the wake are less strong, which could explain that they did not find significant turbulent transport from the center towards the outer region of the wake.

### 5.3 Near wake length

The length of the near wake is defined by the distance downstream from the WT where the transition to the far wake occurs. In Sect. 3 we described an analytical model for estimating the near wake length. The estimated length for the near wake under

stable atmospheric conditions (#206) is $l_{nw} = 2.9$. This length coincides with the development of turbulence intensity in the center of the wake in the downstream direction. The development of the turbulence intensity of the same flight still shows an increase at a distance of $x = 2\,D$, indicating measurements within the near wake, since the downstream position of the turbulence intensity peak is associated with the transition from the near wake in to the far wake (Wu and Porté-Agel, 2012). However, due to the deviation in pattern orientation, the increase in $I$ could also be caused by the lateral position outside the

wake center. Overall, the estimated wake length for the flights considered is between $1.7\,D$ and $2.9\,D$, so it can be assumed that the examined lateral profiles ($x < 1\,D$) are within the near wake. For detailed near wake length studies, extended simultaneous measurement are needed to capture the entire wake characteristics during the measurements.

### 6 Conclusions

In the present experiment, for the first time, a fleet of UAS was successfully deployed to measure the wind flow around a WT.
The simultaneous up- and down-stream measurements show their great potential for detailed WT wake studies, even without any additional instrumentation. The results of velocity and turbulence intensity distributions as well as turbulence spectra and momentum fluxes were discussed and compared with literature. The following statements summarize the study and provide answers to the research questions defined in the introduction:



- – The horizontal velocity deficit and turbulence intensity profile at a downstream distance of $x = 0.5\,D$ under stable to near-neutral conditions clearly outlines a double Gaussian-like distribution, with a lower velocity deficit at the wake center.

- – At the edge of the wake, the lateral momentum fluxes point towards the wake center, while in the inner regions of the wake, the fluxes point toward the edge. In general, the turbulent transport takes place towards the low wind speed region.

- – The blade passing frequency of the tip vortices can be observed under stable atmospheric conditions in the energy spectra. However the peak of the BPF appears to be quite broad due to the "far" downstream distance ($x = 0.5\,D$), the unsteady rotational speed of the WT, and the variation of the inflow wind direction.

- – The downstream recovery of the velocity deficit is captured with distributed measurements in the longitudinal direction. In addition, the downstream evolution of the horizontal velocity deficit profile from the prominent double Gaussian distribution to a single Gaussian distribution is shown.

- – The velocity deficit under unstable atmospheric conditions does not show a double Gaussian shape at downstream distances of $x = 0.5\,D$. In contrast to stable conditions, no decrease of the added turbulence intensity is observed in the wake center which is an indication for very fast mixing and tip vortex breakdown.

The relative positioning of single UAS in the wake in continuously varying inflow conditions is a challenging task. However, in this study, we show that with simultaneous inflow measurements the relative position in the wake can be well estimated. It is a big advantage to operate with multiple UAS simultaneously to capture the spatial extent of the wake. In future field experiments around WTs, the entire fleet of $> 20$ UAS will be used, allowing to capture the entire rotor swept area in more detail, making the exact positioning of a single UAS less relevant. With a larger fleet simultaneous wake profile measurements at different downstream distances, both in vertical and lateral direction will be possible. Additionally, as shown by Wetz et al. (2022), spatial coherence and correlation measurements are possible with the UAS-fleet and can be used for detailed studies of the flow around WTs. In the meantime, the wind algorithm is developed towards a full 3-D wind vector estimator (Wildmann and Wetz, 2022). It could not be used in this study, as a calibration at high wind speeds as observed in this study has not yet been achieved. In future campaigns, it is the goal to retrieve both, the vertical wind component and the vertical momentum fluxes.

Extended experiments are planned in the near future at the Krummendeich Research Wind Farm WiValdi owned by the German Aerospace Center (DLR) (https://windenergy-researchfarm.com/) (Wildmann et al., 2022). In addition to the three WTs, a variety of meteorological instrumentation will be installed at the site, such as four meteorological masts (up to 150 m tall), ground based and nacelle-based wind lidars. A synthesis of this network of different instruments enables detailed research on the flow in WT wakes and its interaction with atmospheric turbulence.





*Data availability.* Wind rose data obtained from the New European Wind Atlas, a free, web-based application developed, owned and oper-
ated by the NEWA Consortium. For additional information see www.neweuropeanwindatlas.eu.

## Appendix A: Turbulence intensity overview for all flights

Besides the velocity deficit, the turbulence intensity is an important quantity in WT wake studies. Therefore, in Fig. A1 the
turbulence intensity $I$ is illustrated and normalized in the same manner as the overview plot of velocity deficit in fig. 4. The
increase of turbulence intensity in the wake region is clearly observable. However, the highest $I$ is expected near the edge
of the wake due to the tip vortices and the continuously changing deflection of the wake due to the variability of the inflow
wind direction. Additionally, the level of freestream turbulence intensity $I_0$ influence the normalized turbulence intensity in
the wake. For example if $I_0$ is small the increase in $I$ in the WT wake can be much higher than for cases with already high
turbulence intensity in the freestream.

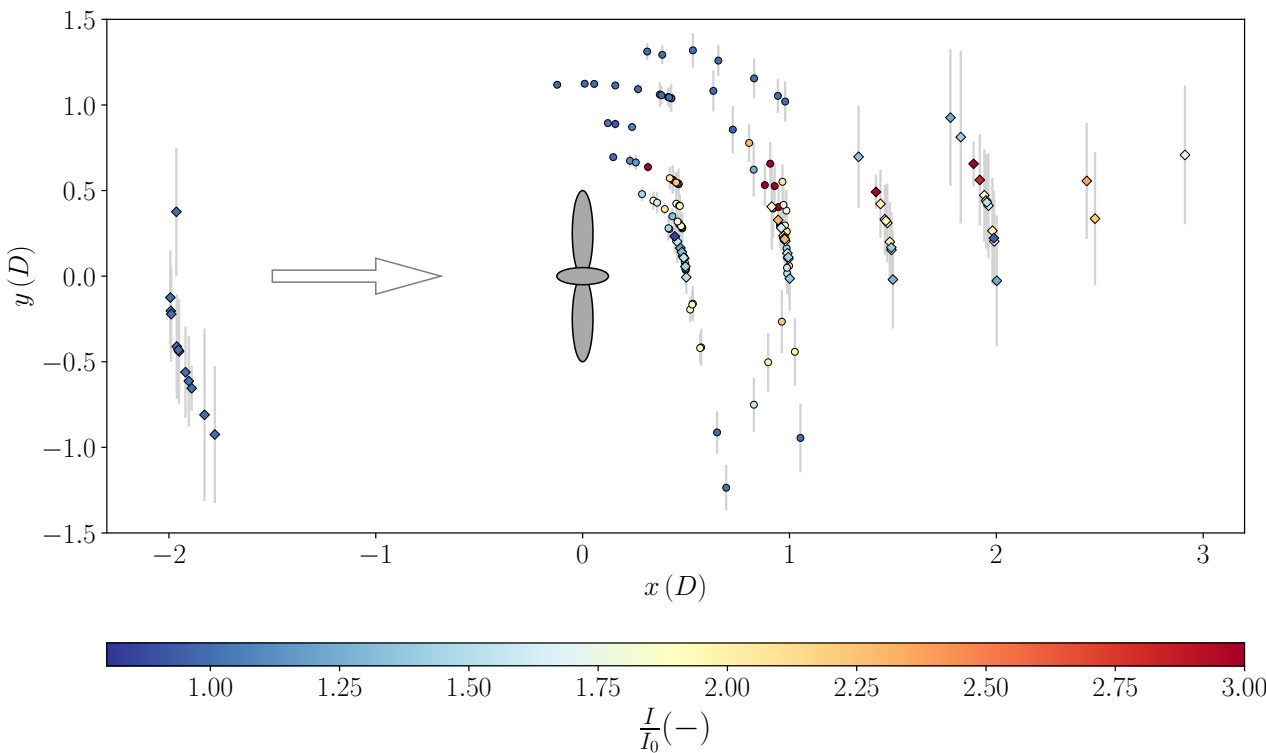

**Figure A1.** Normalized turbulence intensity $I/I_0$ of UAS measurements at locations relative to the reference wind direction.



*Author contributions.* TW wrote the main manuscript and performed the data analysis. The experiment was conducted by TW. NW was
involved in the experiment, contributed to manuscript and prepared parts of Fig. 1.

*Competing interests.* The contact author has declared that none of the authors has any competing interests.

*Acknowledgements.* We thank Almut Alexa, Linus Wrba, Josef Zink and Johannes Kistner for their assistance during the field measurements.
Special thanks goes to EnBW and in particular Carolin Schmitt for the possibility to perform wind measurements in close vicinity to their
WT and for making the WT data accessible for us. Manuel Gutleben internally reviewed the manuscript and we thank him for his valuable
comments.



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
