# Peer review of "Multi-Point In-Situ Measurements of Turbulent Flow in a Wind Turbine Wake and Inflow with a Fleet of UAS"

_Wind Energy Science, 2023_

## Referee Comment (RC2)

Referee's comments to wes-2023-6

**Main comments**

This study represents an important contribution to the topic of experimental wind energy, proposing a new approach to measure the flow around a wind turbine.

The literature is fairly review and the authors cite some of the most relevant studies in the field.

The reader is referred to previous publication for the experimental strategy, although it would be nice to introduce a paragraph discussion experimental uncertainty and data quality control protocols in this manuscript to give a more comprehensive overview of the work.

A missing information appears to be the flight duration, which is relevant for two reasons: *(i)* the longer the flight, the higher the statistical significance; *(ii)* the duration of the flight influences the contribution of mesoscales into the turbulent fluxes. Regarding the latter point, simply providing the standard deviation of wind direction is not sufficient since both turbulence and mesoscale-related unsteadiness contribute to this quantity. It is recommended adding a few lines discussing this point.

A rather theoretical but worthwhile objection can be raised about the use of the "normalized turbulence intensity" in Figures 9b, 12b, and A1. The reason for normalizing the mean wind speed through the incoming freestream lies on the Pi Theorem and the Reynolds analogy assumption, viz. velocity fields observed at different inflows become statistically homogeneous when divided by the incoming mean wind speed, provided that the Reynolds number is high and the geometry of the system stays unchanged. It is hard to justify a normalization of the turbulence intensity based on the freestream value, being this parameter already non-dimensional. The authors themselves normalize the momentum flux in a different way (i.e. dividing by $u_0^2$) and are forced to use the more common added turbulence in several occasions. It is recommended that all the plots using $I/I_0$ are changed to added turbulence.

The paper can be in the Referee's opinion accepted if these minor comments are addressed.

**Specific comments:**

Line 4: it would be useful expand the acronym "UAS" at the first usage.

Lines 39: the near wake is more correctly affected by the local aerodynamic forces on the rotor, so it may be better to say "[…] is closely related to the design and operation of the WT".

Line 40: same as same comment, it should be changed to "detailed design and operation of the WT".

Line 80: "conduct" seems incorrect.

Line 147: can you please clarify in the manuscript if the orientation of the wind turbine is assessed visually or from the SCADA data?

Line 167: "standard deviation of the streamwise velocity" should be indicated explicitly at this line and omitted at line 170.

Lines 175-176: can you provide a reference for the choice of the lapse rate thresholds?

Line 180: if $\Omega$ is the rotational frequency (i.e. revolutions per second) the angular velocity is $\omega = \Omega \cdot 2\,\pi$, not divided as in the formula provided. Also, please define $\Omega$ explicitly as rotational frequency, not "speed".

Equation 7: please clarify how the $c_T$ is estimated, since it is generally not measured by the SCADA.

Line 213-214: it is not clear how the dots in Fig. 4 can be aligned with "wind direction and the WT" if there is a significant yaw misalignment most of the times. Please clarify.

Lines 221-222: the reason for a more likely counter-clockwise misalignment of the UAS is not obvious, please expand on this point.

Figure 5: independently from the UAS measurements, the nacelle orientation shows negative bias with respect to the wind direction from SCADA. Please comment on this difference.

Lines 290-291: please specify that the peak prominence is smeared by the dissipation of tip vortices that are diffused over 0.5 D.

Figure 7: the calculation of the boundary of the grey band around the BPF is not clear. Explain better or remove.

Line 310: the statement "vortices remain stable for longer distances" to explain turbulent mixing may be confusing considering the previous discussion on tip vortices. Please revise.

Lines 377-378: why convective conditions are more frequent in complex terrain? Please clarify or remove.

Lines 388-390: please expand on the lower added TI in the wake center for stable conditions, which seems to contradict the higher momentum fluxes.

---

## Author Comment (AC1)

**Multi-Point In-Situ Measurements of Turbulent Flow in a Wind Turbine Wake and Inflow with a Fleet of UAS**

Tamino Wetz[1] and Norman Wildmann[1]

[1]Deutsches Zentrum für Luft- und Raumfahrt e.V., Institut für Physik der Atmosphäre, Oberpfaffenhofen, Germany

**Correspondence:** Tamino Wetz (tamino.wetz@dlr.de)

**1 Review response**

We want to thank the anonymous reviewer and Stefano Letizia for their valuable feedback and valid points of criticism to our manuscript.

**1.1 Review Comment 1**

**1.1.1 RC1, General Comments**

*This study employs a fleet of UAS to measure the flow around a 2 MW wind turbine under various configurations and atmospheric conditions. The velocity and turbulence intensity in the wake are quantified. Exciting and convincing results characterizing the momentum fluxes in the wake and the effect of atmospheric stability on the wake shape are presented. Overall, the manuscript is well-written and provides novel contributions to the field of in-situ wind turbine wake measurements. The study is well within the scope of Wind Energy Science, and I strongly support its publication.*

We thank the reviewer for their positive feedback and support as well as their constructive comments to improve the manuscript.

**1.1.2 RC1, Specific Comments**

1. *Table 1 could benefit from the inclusion of some additional information. In the "pattern" column, it would be helpful to know at which longitudinal distance the lateral patterns were located for each case. What was the duration of each flight? Also, adding the lapse rate for each case would provide a sense of how strongly stable or convective the boundary layer was during each measurement. Finally, the region of operation of the turbine (i.e., above or below rated power) will substantially affect the strength of the wake at any given time, so this must be included as well.*

   We include the longitudinal distance of the lateral pattern as well as the lapse rate for each case. The rated power is reached at a wind speed of $12.5 \, \mathrm{m \, s^{-1}}$ so the provided inflow wind speed gives an indicator in which stage the WT is operating. For most of the present flight cases, the WT is operating below rated power, only for flights $\# \, 206$ and $\# \, 207$ the WT operated near rated power. A statement is added to the manuscript.

2. *Page 13-14: The discussion of the tip vortex measurements is not very strong. I agree that all of the factors listed will weaken the peak in the frequency spectrum. However, is the signature of the tip vortices visible in any of the time series taken at the wake edge location, even for just a short time? Is this what is meant by the line "approximately seven tip vortices could be observed between the measurement position and the WT"? Especially because tip vortex breakdown is discussed throughout the manuscript, it would be beneficial to elaborate a bit here.*

Thank you for the comment. By seven observable tip vortices, we meant that if you take a snapshot of the wake flow, you can theoretically see, based on the revolution of the WT and the distance of the measurement, 7 tip vortices on a line between the measurement position and the WT. With this, we want to outline the travel history of the tip vortex. We clarified this statement also in the manuscript.

The signature of the tip vortices in a time series of a single point measurement is difficult to capture, since the signature strongly depends on the position relative to the center of the vortices. However, the time series of the lateral velocity $v$ of the relevant UAS ($\overline{y} = 0.55\,D$) measurement of flight $\#\,606$ shows in some periods the signature of the tip vortex. One segment of the time series of the lateral velocity component $v$ where the signature is clearly visible, is shown in Fig. 1a, due to confidential reasons the labels on the y-axis have been removed. The signature of the tip vortex at hub height is characterized by a strong increase in lateral velocity followed by an abrupt change towards the opposite lateral direction and ending back in the ambient lateral wind velocity. Or, depending on the relative measurement position concerning the vortex center, the vortex can only cause a strong increase or decrease in the streamwise velocity without influencing the lateral velocity. This discussion is added to the manuscript and the figure, including additional descriptions, is added to the appendix. Additional discussions on the basis of the present dataset would be quite speculative, thus we limited the discussion on the tip vortices at this stage, and detailed experiments on this topic are necessary and planned in order to extend the discussion.

3. *Page 15: It would be nice to have a discussion of 1-2 more examples from Fig. 9a. For example, in case $\#\,708$ the velocity shows a strong increase between 1D and 2D. Do you think this is because the large value of $\gamma$ is causing the centerline of the wake to be deflected away from the measurement location? Or does the large value of Ix indicate that the wake is already starting to recover?*

A discussion of flight case $\#\,708$ is added to the manuscript. Both, the misalignment of the WT and the deviation of the pattern orientation caused the measurement to be outside the centerline, which explains the increase in wind speed in all cases. However, in this specific case and regarding the turbulence intensity of the same measurement (Fig 9b) a decrease can be observed towards x=2 D which could indicate that the wake recovery has already started, which also leads to an increase in wind speed at this longitudinal distance. This could be plausible in this case due to the high level of ambient turbulence, which supports the wake recovery.

4. *Page 17, lines 361-362: If future researchers are to use this UAS method to measure wind turbine wake flows, it would be helpful to have some elaboration on the difficulties. What do you think are the sources of uncertainty in the lateral velocity measurements and how could they be addressed in future studies? Do you think there could be vertical wake*

[Figure]

**Figure 1.** Time-series of the lateral velocity component a), wind direction in b) and lateral position c) of a lateral flight pattern in stable conditions (fl. # 606). The lateral positions are calculated using the wind direction of reference UAS measurement, which is shown in the middle figure. The black bar in c) indicates the lateral position of the WT.

*deflection caused by the terrain? Or is this due to some uncertainty in the lateral velocity measurements themselves or the positioning of the UAS? This information could help future researchers decide whether this method is appropriate for their investigations. This elaboration could also be added to the conclusion section of the manuscript.*

In the conclusion, we already included sources of uncertainty for the measurements. However, we extend the statement about the difficulty of far-wake measurements at this point and include it also in the conclusion. The complex terrain on one hand can cause additional variability in the inflow wind direction, which leads to large lateral deflections of the wake and makes the positioning of single point measurements in the wake challenging. On the other hand, the present complex terrain could also cause vertical deflection due to the significant slope to the west of the WT, which could also affect the vertical position of the WT wake. Strong terrain effects were for example shown in the Perdigao experiment with dual-Doppler lidar measurements (Wildmann et al., 2018). However, in that case, the WT was directly on a mountain ridge with slopes starting only few tens of meters up and downstream of the WT, in our case, there is no steep change

in terrain downstream and the escarpment upstream is more than two rotor diameters away. A similar sight has been investigated at the Swabian Alp, which shows that flow inclination quickly recovers after the escarpment (Wildmann et al., 2017). Since the lateral velocity components are comparably small, the uncertainty for the velocity measurements are more crucial for this component.

5. *Figures 9 and 10: For the longitudinal flight patterns, did the authors consider applying conditional averaging? Since the wind direction is changing, you could take the average of all velocities that are measured when the instantaneous misalignment angle $\beta$ is less than a certain value (see references below).*

   From the complete dataset, we already only present flight cases with a pattern misalignment angle $\beta$ of less than 20 degrees. We appreciate the idea of grouping the data depending on the misalignment angle $\beta$. However, for the remaining flight cases, the number of flight cases is too low to obtain a significant statement by grouping the flight cases. Therefore, we decide to present the flight cases separately without grouping, so that the individual flight cases are transparently visible.

6. *Page 20: A couple other field studies have observed the double Gaussian shape of the near wake. These should be included for completeness:*

   – *Abraham, A., Dasari, T., and Hong, J. (2019). Effect of turbine nacelle and tower on the near wake of a utility-scale wind turbine. Journal of Wind Engineering and Industrial Aerodynamics, 193, 103981.*

   – *Keane, A. (2021). Advancement of an analytical double-Gaussian full wind turbine wake model. Renewable Energy, 171, 687-708.*

   We appreciated the additional literature for field studies and included both in the manuscript.

**1.1.3   RC1, Technical corrections**

1. *Why are $\Delta x$ and $\Delta y$ used rather than x and y throughout the manuscript? If they are different, make sure they are clearly defined.* The manuscript is corrected towards x and y, since the origin of the coordinate system is placed at the turbine hub, so the distance to the WT equals the coordinate points.

2. *Figure 1: The numbers on the elevation contours are not clearly visible.* The position and size of the numbers on the elevation contours are adjusted.

3. *Page 7, lines 175-176: "laps" should be "lapse".* Corrected.

4. *Figure 4 legend: There is no black bar in the figure.* Corrected.

5. *Figures 4, 8-14, and A1: Please make the symbols bigger so the different cases can be differentiated more easily.* All figures are updated accordingly in the manuscript.

6. *Figure 7: Does the dotted line represent k-5/3? Also based on the caption, the y-axis label should be Su.* Thank you for the comment. We corrected the dotted line label to $k = -5/3$ and the axis label.

7. *Page 22, line 435: I believe the authors are referring to a different case, as $\#$ 206 occurs under unstable atmospheric conditions per Table 1.* Corrected.

**References**

Wildmann, N., Bernard, S., and Bange, J.: Measuring the local wind field at an escarpment using small remotely-piloted aircraft, Renewable Energy, 103, 613 – 619, https://doi.org/https://doi.org/10.1016/j.renene.2016.10.073, 2017.

Wildmann, N., Kigle, S., and Gerz, T.: Coplanar lidar measurement of a single wind energy converter wake in distinct atmospheric stability regimes at the Perdigão 2017 experiment, J. Phys.: Conf. Ser., 1037, 052 006, https://doi.org/10.1088/1742-6596/1037/5/052006, 2018.

**Relevant changes to the manuscript**

We list here the relevant changes to the manuscript:

1. Introduction:

   – Text modifications in response to referee comments.

2. Experiment:

   – Text modifications in response to referee comments.

   – Fig. 3 is modified in response to referee comments.

3. Methods:

   – Text modifications in response to referee comments.

   – Table 1 is modified in response to referee comments.

4. Results:

   – Text modifications in response to referee comments.

   – Fig. 4 is modified in response to referee comments.

   – Fig. 7 is modified in response to referee comments.

   – Fig. 8 is modified in response to referee comments.

   – Fig. 9 is modified in response to referee comments.

   – Fig. 10 is modified in response to referee comments.

   – Fig. 11 is modified in response to referee comments.

   – Fig. 12 is modified in response to referee comments.

   – Fig. 13 is already included now in Fig. 12 and therefore removed in response to referee comments.

   – Fig. 14 is modified in response to referee comments.

5. Discussion:

   – Text modifications in response to referee comments.

6. Conclusion:

   – Text modifications in response to referee comments.

130 7. Appendix:

   – Fig. A1 is modified in response to referee comments.

   – Fig. B1 is added in response to referee comments.

---

## Author Comment (AC2)

**Multi-Point In-Situ Measurements of Turbulent Flow in a Wind Turbine Wake and Inflow with a Fleet of UAS**

Tamino Wetz[1] and Norman Wildmann[1]

[1]Deutsches Zentrum für Luft- und Raumfahrt e.V., Institut für Physik der Atmosphäre, Oberpfaffenhofen, Germany

**Correspondence:** Tamino Wetz (tamino.wetz@dlr.de)

**1  Review response**

We want to thank the anonymous reviewer and Stefano Letizia for their valuable feedback and valid points of criticism to our manuscript.

**1.1  Review Comment 2**

**1.1.1  RC2, General Comments**

*This study represents an important contribution to the topic of experimental wind energy, proposing a new approach to measure the flow around a wind turbine.*

*The literature is fairly review and the authors cite some of the most relevant studies in the field.*

*The reader is referred to previous publication for the experimental strategy, although it would be nice to introduce a paragraph discussion experimental uncertainty and data quality control protocols in this manuscript to give a more comprehensive overview of the work.*

*A missing information appears to be the flight duration, which is relevant for two reasons: (i) the longer the flight, the higher the statistical significance; (ii) the duration of the flight influences the contribution of mesoscales into the turbulent fluxes. Regarding the latter point, simply providing the standard deviation of wind direction is not sufficient since both turbulence and mesoscale-related unsteadiness contribute to this quantity. It is recommended adding a few lines discussing this point. A rather theoretical but worthwhile objection can be raised about the use of the "normalized turbulence intensity" in Figures 9b, 12b, and A1. The reason for normalizing the mean wind speed through the incoming freestream lies on the Pi Theorem and the Reynolds analogy assumption, viz. velocity fields observed at different inflows become statistically homogeneous when divided by the incoming mean wind speed, provided that the Reynolds number is high and the geometry of the system stays unchanged. It is hard to justify a normalization of the turbulence intensity based on the freestream value, being this parameter already non-dimensional. The authors themselves normalize the momentum flux in a different way (i.e. dividing by $u_0^2$) and are forced to use the more common added turbulence in several occasions. It is recommended that all the plots using $I/I_0$ are changed to added turbulence.*

*The paper can be in the Referee's opinion accepted if these minor comments are addressed.*

25 We agree that the chapter about the measurement system is short. The accuracy of the measurement is additionally provided in the description of the experimental setup. However, the focus of that study is on the application of the measurement system, as you mentioned, detailed information can be found in the references and is out of scope here.

The flight duration is outlined more prominently in the manuscript for clarification. It is rather the first issue of too short flights for statistical significance, which is more relevance for our measurements with 10...12 minutes flight time.

30 We agree that both turbulence and mesoscale-related unsteadiness contribute to the standard deviation of the wind direction. Since we are limited to 10 min wind data mesoscale-related unsteadiness cannot be captured entirely. However, in the IEC-standard 61400-12-1, the calculation of the turbulence intensity and the mean wind speed is also based on 10 min data.

We agree that the normalization of the turbulence intensity does not follow the Pi theorem since it is already dimensionless. As the added turbulence is more common in literature, we changed the mentioned figures from normalized turbulence intensity to

35 added turbulence intensity and removed the resulting redundant figure.

**1.1.2 RC2, Specific Comments**

1. *Line 4: it would be useful expand the acronym "UAS" at the first usage.*
   We agree and have revised the manuscript accordingly.

40 2. *Lines 39: the near wake is more correctly affected by the local aerodynamic forces on the rotor, so it may be better to say "[...] is closely related to the design and operation of the WT".*
   We agree and have revised the manuscript accordingly.

3. *Line 40: same as same comment, it should be changed to "detailed design and operation of the WT".*
   We agree and have revised the manuscript accordingly.

45 4. *Line 80: "conduct" seems incorrect.*
   We agree and have revised the manuscript accordingly.

5. *Line 147: can you please clarify in the manuscript if the orientation of the wind turbine is assessed visually or from the SCADA data?*
   We have revised the manuscript accordingly and added a statement. The orientation of the WT is obtained from the
50 SCADA data and supported visually if short-term changes are observed.

6. *Line 167: "standard deviation of the streamwise velocity" should be indicated explicitly at this line and omitted at line 170.*
   We agree and have revised the manuscript accordingly.

7. *Lines 175-176: can you provide a reference for the choice of the lapse rate thresholds?*
55 In general, if the lapse rate is positive, the atmosphere is statistically stable and for negative values, it is unstable.

However, the threshold of 0.5 to differentiate between neutral and stable/unstable conditions is found by Mohan (1998). The reference is added to the manuscript.

8. *Line 180: if $\Omega$ is the rotational frequency (i.e. revolutions per second) the angular velocity is $\omega = \Omega * 2\pi$, not divided as in the formula provided. Also, please define $\Omega$ explicitly as rotational frequency, not "speed".*

We agree and have revised the manuscript accordingly.

9. *Equation 7: please clarify how the $c_T$ is estimated, since it is generally not measured by the SCADA.*

The $c_T$ is given by the WT operator and details can not be shared due to confidentiality reasons.

10. *Line 213-214: it is not clear how the dots in Fig. 4 can be aligned with "wind direction and the WT" if there is a significant yaw misalignment most of the times. Please clarify.*

We include an additional statement for clarification. The dots are aligned with the wind direction, and the coordinate origin is locked in the center of the WT but is independent of the orientation of the WT.

11. *Lines 221-222: the reason for a more likely counter-clockwise misalignment of the UAS is not obvious, please expand on this point.*

By this point, we have only described the fact that there is a obvious trend in the plot which results from the alignment of the pattern towards the inflow wind direction and the orientation of the nacelle. As outlined in the manuscript, there is a trend of a yaw misalignment of the WT, which then also causes the orientation of the UAS to be misalignment with regard to the inflow wind direction. Reasons for the misalignments are given in Section 4.2. The statement is adjusted in the manuscript.

12. *Figure 5: independently from the UAS measurements, the nacelle orientation shows negative bias with respect to the wind direction from SCADA. Please comment on this difference.*

This can only be answered by the operator of the WT and due to confidentiality reasons, we can not comment on this topic.

13. *Lines 290-291: please specify that the peak prominence is smeared by the dissipation of tip vortices that are diffused over 0.5 D.*

An additional specification is added to the manuscript.

14. *Figure 7: the calculation of the boundary of the grey band around the BPF is not clear. Explain better or remove.*

We include an additional explanation for the calculation of the frequency band around the BPF, which is calculated from the maximum and minimum rotation frequencies of the WT measured during the considered time period.

15. *Line 310: the statement "vortices remain stable for longer distances" to explain turbulent mixing may be confusing considering the previous discussion on tip vortices. Please revise.*

We changed the statement and only argue with less-pronounced turbulence mixing.

16. *Lines 377-378: why convective conditions are more frequent in complex terrain? Please clarify or remove.*

    We agree and removed the statement.

17. *Lines 388-390: please expand on the lower added TI in the wake center for stable conditions, which seems to contradict the higher momentum fluxes.*

    In our opinion, less added TI in the wake center does not automatically contradict higher momentum fluxes in the center part of the wake. Rather, the inhomogeneous velocity deficit profile in stable stratifications leads to an increased momentum flux in the wake center.

**References**

95  Mohan, M.: Analysis of various schemes for the estimation of atmospheric stability classification, Atmospheric Environment, 32, 3775–3781, https://doi.org/10.1016/s1352-2310(98)00109-5, 1998.

**Relevant changes to the manuscript**

We list here the relevant changes to the manuscript:

1. Introduction:

100      – Text modifications in response to referee comments.

2. Experiment:

      – Text modifications in response to referee comments.
      – Fig. 3 is modified in response to referee comments.

3. Methods:

105      – Text modifications in response to referee comments.
      – Table 1 is modified in response to referee comments.

4. Results:

      – Text modifications in response to referee comments.
      – Fig. 4 is modified in response to referee comments.
110      – Fig. 7 is modified in response to referee comments.
      – Fig. 8 is modified in response to referee comments.
      – Fig. 9 is modified in response to referee comments.
      – Fig. 10 is modified in response to referee comments.
      – Fig. 11 is modified in response to referee comments.
115      – Fig. 12 is modified in response to referee comments.
      – Fig. 13 is already included now in Fig. 12 and therefore removed in response to referee comments.
      – Fig. 14 is modified in response to referee comments.

5. Discussion:

      – Text modifications in response to referee comments.

120  6. Conclusion:

   – Text modifications in response to referee comments.

7. Appendix:

   – Fig. A1 is modified in response to referee comments.

   – Fig. B1 is added in response to referee comments.